# Impact of IASI thermal infrared measurements on global ozone reanalyses

Emanuele Emili[1] and Mohammad El Aabaribaoune[1]

[1]CECI, Université de Toulouse, Cerfacs, CNRS, Toulouse, France

**Correspondence:** E. Emili (emili@cerfacs.fr)

**Abstract.** The information content of thermal infrared measurements for tropospheric $O_3$ estimation has already been well demonstrated. However, the impact of such measurements to constrain modelled ozone distributions within global assimilation systems is not yet unequivocal. A new tropospheric $O_3$ reanalysis is computed for the year 2010 by means of assimilating measurements from the Infrared Atmospheric Sounding Interferometer (IASI) within the MOCAGE chemical transport model.
The objective is to evaluate the impact of recent methodological improvements of the data assimilation scheme on the $O_3$ distribution. The new $O_3$ reanalysis (named IASI-r) and its precursor (IASI-a) have been validated against ozonesondes, compared to independent estimations of tropospheric $O_3$ and to results from two state-of-the-art products based on detailed tropospheric chemistry (GEOS-CCM and CAMS reanalysis). The main difference between IASI-r and the former IASI-a concerns the treatment of IASI observations, with radiances being assimilated directly in IASI-r instead of intermediate Level 2 $O_3$ retrievals. IASI-r is found to correct major issues of IASI-a, i.e. the neutral or negative impact of IASI assimilation in the extra-tropics and the presence of residual biases in the tropics. IASI-r compares also relatively well to the CAMS reanalysis, which is based on more comprehensive chemical mechanism and the assimilation of several UV and microwave measurements.

## 1  Introduction

Tropospheric ozone ($O_3$) contributes to global warming with a net positive effect in the Upper Troposphere - Lower Stratosphere (UTLS) at tropical and sub-tropical latitudes (Stevenson et al., 2013). Tropospheric $O_3$ evolution is driven by both natural and anthropogenic processes such as photochemical production from primary pollutants, convection, long range transport and stratosphere-troposphere exchanges (Young et al., 2013). Furthermore, natural atmospheric oscillations like the El Niño-Southern Oscillation (ENSO) can contribute significantly to the variability of the $O_3$ concentration in the tropical UTLS (Ziemke et al., 2015).

The complexity of $O_3$ processes (sources, sinks and transport patterns) and the scarcity of direct $O_3$ measurements in the UTLS (Cooper et al., 2014) make the use of atmospheric composition models necessary for the estimation of the $O_3$ radiative forcing (Gauss et al., 2006). Recent model inter-comparison exercises such as the Atmospheric Chemistry and Climate Model Intercomparison Project (ACCMIP, Lamarque et al. (2013)) estimated the present day $O_3$ radiative forcing uncertainty to be 30 % (Stevenson et al., 2013), due to uncertainties in $O_3$ modelling.

Satellite measurements provide global and independent estimates of $O_3$, although with limited vertical information. An estimation of tropospheric $O_3$ columns can be obtained by combining stratospheric profiles from microwave limb sounders with total $O_3$ columns from Ultra-Violet (UV) nadir sounders. For example, Ziemke et al. (2006) combined retrievals from the Ozone Monitoring Instrument (OMI) and from the Microwave Limb Sounder (MLS) onboard AURA satellite to provide a Tropospheric Ozone Column (TOC) product, which will be called hereafter OMI-MLS. Thermal sounders like the Tropospheric Emission Spectrometer (TES, Bowman et al. (2006)) or the Infrared Atmospheric Sounding Interferometer (IASI, Clerbaux et al. (2009); Hilton et al. (2012)) add complementary information to UV and limb sounders thanks to a peak of sensitivity in the UTLS region within the 10 $\mu$m spectral region. Another advantage of thermal sounders is the possibility to retrieve $O_3$ also during night-time, thus providing observations during the polar nights. Recently, there have been growing efforts to comparatively evaluate tropospheric $O_3$ retrievals from UV and IR measurements (Gaudel et al., 2018) in the perspective of present $O_3$ variability and trends. These studies highlighted some inconsistencies between UV and IR retrievals in terms of $O_3$ columns and observed trends, that are currently being addressed.

Satellite measurements are also assimilated routinely in global chemistry models (Bocquet et al., 2015) to either improve chemical forecasts (Flemming et al., 2015) or to produce more accurate reanalyses of the past atmospheric composition (Inness et al., 2019). However, due to the difficulties to harmonise more recent IR products with well established UV ones, chemical assimilation systems still rely mostly on UV or limb $O_3$ measurements (Van Peet et al., 2018; Inness et al., 2019). Because of the weak sensitivity of UV measurements to the troposphere, tropospheric $O_3$ is thus constrained indirectly through the simultaneous fit to total and stratospheric columns. An exception is represented by the reanalysis of Miyazaki et al. (2015), which assimilated tropospheric TES $O_3$ profiles. However, the reduced availability of TES data after 2010 impeded the production of an $O_3$ reanalysis constrained by IR measurements for the entire period. Huijnen et al. (2020) compared the accuracy of the tropospheric $O_3$ reanalyses of Inness et al. (2019) and Miyazaki et al. (2015) against radio-soundings and found similar accuracies for both. The IASI mission provides uninterrupted measurements from 2008, with three sensors flying today (Metop A, B and C) and plans for longterm continuation of operations in the next decade (IASI-NG). Although IASI $O_3$ observations might be very valuable for long reanalyses, they have not yet been assimilated within any of the currently available chemical reanalysis.

Global assimilation of IASI $O_3$ products have been examined in a number of previous studies. Massart et al. (2009) was the first to assimilate IASI total $O_3$ columns in a Chemical Transport Model (CTM) and highlighted the need to employ averaging kernels to correctly weight the instrument sensitivity. Emili et al. (2014) examined the assimilation of tropospheric columns (240-1000 hPa) using retrievals kernels and found a positive impact at tropical latitudes but slightly negative at mid and high latitudes. Peiro et al. (2018) used the same methodology as Emili et al. (2014) to compute tropospheric $O_3$ analyses in the tropics to study the $O_3$ ENSO signal. All these studies performed a bias correction prior to the assimilation of retrieved quantities (also named Level 2 or L2 products) to avoid increased biases in the $O_3$ reanalyses. The empirical nature of the bias correction procedure could not ensure positive (or neutral) impact of IR assimilation at all latitudes and vertical levels. More recently, Emili et al. (2019) evaluated the direct assimilation of Level 1 (L1) radiances from IASI to overcome some of the difficulties encountered before and found that more variability can be extracted from IASI spectra when assimilating

directly the radiances. Barret et al. (2020) confirmed that the use of a dynamical a-priori based on the tropopause height improved significantly $O_3$ retrievals at southern mid latitudes. Finally Aabaribaoune et al. (2020) improved the L1 assimilation scheme by employing a more realistic observation error covariance for the L1 radiances, which reduced residual stratospheric biases. However, both the analyses of Emili et al. (2019) and Aabaribaoune et al. (2020) were limited to a single summer

5    month in 2010 and could not draw conclusions on the capacity of IR assimilation to reproduce the seasonal $O_3$ variability or trends in the extra-tropics. Note that the assimilation of IASI radiances sensitive to $O_3$ was also investigated in the framework of Numerical Weather Prediction (NWP) by Dragani and Mcnally (2013), and more recently Coopmann et al. (2018). $O_3$ information from IR sounders could in fact improve NWP through the tracer effect (Dragani and Mcnally, 2013) or thanks to increased temperature and humidity information content (Coopmann et al., 2020). However, some difficulties still persist with

respect to the correction of biases and the appearance of drifts related to the variational bias correction (Dragani and Mcnally, 2013). To conclude, there is a strong need to further assess the benefits of IASI IR observations for UTLS $O_3$ analyses both for chemical and meteorological applications.

      The objective of this paper is to evaluate a new $O_3$ reanalysis during a full year (2010), based on the assimilation of IASI measurements within the CTM MOCAGE (Josse et al., 2004). This new reanalysis profits from all methodological improve-

ments achieved in the recent years concerning IASI $O_3$ assimilation. In particular, the major difference with the multi-year $O_3$ reanalysis of Peiro et al. (2018) (hereafter called IASI-a) is that the IASI L1 spectrum is now assimilated directly and without performing any empirical bias correction. In this study we keep the simpler description of $O_3$ based on linearised chemistry (Cariolle and Teyssedre, 2007) already used in previous global reanalyses (Massart et al., 2009; Emili et al., 2014; Peiro et al., 2018) to assess the improvements of the assimilation scheme alone. On top of the usual validation against ozone-sondes mea-

surements, the obtained $O_3$ fields will be compared to IASI-a, to output from a climate-chemistry model used in ACCMIP (GEOS-CCM) and to the CAMS atmospheric composition reanalysis (CAMSRA, Inness et al. (2019)). We will thus verify the added value of IR assimilation to map the $O_3$ 3D distribution with respect to existent state-of-the-art techniques based either on full chemistry modelling or combination of modelling and satellite data assimilation (non IR). Results with the tested configuration might also be of interest for NWP models, that are generally based on $O_3$ linearised schemes (Han and McNally,

2010) for reasons of computational efficiency. In the context of ongoing efforts to include more detailed chemical modelling within NWP, this study can thus provide further insights on the tradeoff between adding more complexity to the model and better exploiting current and future satellite measurements.

      The paper is structured as follows. Section 2 resumes the $O_3$ modelling and reanalysis datasets that were already publicly available prior to this work and that are used here for comparison. The measurements that are assimilated to produce the new

reanalysis and those that are used for independent validation are detailed in Sec. 3. The configuration of the chemistry transport model and the setup of the assimilation experiment is described in Sec. 4. The results section (Sec. 5) is split in four subsections: the first (Sec. 5.1) is focused on the validation of the new reanalysis against ozonesondes and with respect to IASI-a, the second, third and fourth subsections (5.2, 5.3, 5.4) provide some further comparisons with tropospheric $O_3$ retrievals and the modelling and assimilation datasets of ACCMIP and CAMSRA. Conclusions and perspectives of this work are summarised in the last

section.

## 2 Previous modeling and assimilation experiments

### 2.1 GEOS-CCM

The Goddard Earth Observing System (GEOS) version 5.0 is a general circulation model developed by the National Aeronautics and Space Administration (NASA) that includes an interactive chemistry module for coupled Chemistry-Climate Modelling (CCM). It is one of the models with detailed stratospheric and tropospheric chemistry schemes that took part in the ACCMIP exercise (Lamarque et al., 2013) and was used to estimate present and future $O_3$ radiative forcing (Stevenson et al., 2013). It also has one of the lowest tropospheric $O_3$ bias amongst the ACCMIP historical simulations (Young et al., 2013) and Oman et al. (2011) showed that it reproduces well the tropospheric $O_3$ variability linked to ENSO. In the context of this study, we consider GEOS-CCM representing state-of-the-art climate-chemistry simulations (without data assimilation). We downloaded monthly $O_3$ fields for 2010 from the historical climate ACCMIP simulation publicly available on the CEDA archive (http://archive.ceda.ac.uk, last access 15/06/2020). Details on the setup used for this simulation can be found in Lamarque et al. (2013).

### 2.2 CAMSRA

The chemical reanalysis provided by the Copernicus Atmosphere Monitoring Services (CAMS) is based on the Integrated Forecast System (IFS) general circulation model developed at ECMWF, which integrates detailed tropospheric chemistry and aerosols schemes (Flemming et al., 2015). Within IFS, the NWP data assimilation scheme was extended to assimilate satellite measurements of trace gases and aerosols. In the latest version of the CAMS reanalysis of the atmospheric composition (CAMSRA, Inness et al. (2019)) measurements of $O_3$, CO, $NO_2$ and Aerosols Optical Depth (AOD) are assimilated from a variety of satellites. Concerning $O_3$, no tropospheric measurements are currently assimilated in CAMSRA. Thus, tropospheric $O_3$ is constrained indirectly, either through the combination of total and stratospheric $O_3$ observations or through the assimilation of its chemical precursors (e.g. $NO_2$). Extensive validation (Inness et al., 2019; Huijnen et al., 2020) showed that biases of the CAMSRA $O_3$ reanalysis with respect to ozonesondes are generally lower than 10% in the troposphere, compared to values of about 15% for the ACCMIP free model simulations. Hence, CAMSRA represents a state-of-the-art $O_3$ reanalysis based both on comprehensive chemical modelling and most of available satellite observations (except for IR sounders). CAMSRA monthly $O_3$ fields have been downloaded from the CAMS data store (https://ads.atmosphere.copernicus.eu, last access 16/06/2020).

### 2.3 IASI-a

The IASI-a global $O_3$ reanalysis (Peiro et al., 2018) is an outcome of several projects focused on the exploitation of satellite data for atmospheric analyses, that began with the Assimilation of Envisat Data (ASSET) project (Lahoz et al., 2007; Geer et al., 2006). In recent years, a particular effort has been dedicated to hyperspectral IR measurements from IASI, which provide a wealth of information for tropospheric chemistry, but are not yet assimilated in most operational centres. The IASI-a $O_3$ analysis discussed by Peiro et al. (2018) is based on a linearised $O_3$ chemistry configuration of the MOdéle pour la Chimie

Á Grande Echelle (MOCAGE, Josse et al. (2004); Cariolle and Teyssedre (2007)) and on the joint assimilation of IASI tropospheric columns and stratospheric profiles from the Microwave Limb Sounder (Emili et al., 2014). SOFRID $O_3$ retrievals (Barret et al., 2011), which are based on a variational retrieval scheme and the radiative transfer model RTTOV v9 (Matricardi, 2009), have been assimilated within IASI-a. The biases of the IASI-a analysis are found to be smaller than 15% in the tropical free troposphere (Peiro et al., 2018) suggesting that dense and frequent satellite observations such as IASI ones can partially make up for missing tropospheric chemistry in the model. However, no added value of IASI assimilation was found in the extra-tropics, and even a degradation in the Southern hemisphere mid-latitudes (Emili et al., 2014). Nevertheless, the IASI-a reanalysis represents a reference to evaluate potential improvements with respect to the assimilation of IASI measurements.

## 3 Measurements

In this study we assimilate infrared spectra from the IASI sensor onboard Metop-A satellite (Clerbaux et al., 2009) and stratospheric $O_3$ profiles retrieved from MLS limb measurements (Froidevaux et al., 2008).

IASI spectral measurements (L1c data) contain calibrated and geolocalized spectra at 0.5 $cm^{-1}$ spectral resolution, i.e. 8461 radiance values for each ground-pixel and a footprint of 12 km. Historical L1c data granules have been downloaded from the EUMETSAT Earth Observation data portal (https://eoportal.eumetsat.int, last access 17/06/2020) in NETCDF format. Data files contain also collocated land mask and cloud fraction values, obtained from the Advanced Very High Resolution Radiometer (AVHRR) measurements, also onboard Metop. Since the AVHRR cloud mask is not available before 20/05/2010 we also downloaded the IASI L2 cloud products provided by EUMETSAT, which are based on a combination of multiple cloud detection algorithms (EUMETSAT, 2017).

The MLS V4.2 product (Livesey and al., 2018) contains retrieved $O_3$ profiles on 55 pressure levels ranging from 316 to 0.001 hPa with corresponding retrieval errors. Data have been downloaded from the Goddard Earth Sciences Data and Information Services Center (GES DISC) web portal (https://disc.gsfc.nasa.gov, last access 17/06/2020). For a more detailed discussion on the accuracy of MLS retrievals for $O_3$ analyses please refer to Emili et al. (2019) or Errera et al. (2019).

Ozonesonde profiles and tropospheric columns derived from OMI-MLS observations (Ziemke et al., 2006) are used to compare the $O_3$ simulations to independent measurements. OMI-MLS tropospheric $O_3$ columns have already been used in the past to evaluate the tropical $O_3$ variability and compared to atmospheric chemistry models (Young et al., 2017; Ziemke et al., 2019). Ozonesonde data have been downloaded from the World Ozone and Ultraviolet Radiation Data Centre (WOUDC, http://www.woudc.org, last access 17/06/2020) and OMI-MLS tropospheric $O_3$ from the GODDARD tropospheric ozone archive (https://acd-ext.gsfc.nasa.gov, last access 18/06/2020). To compute model tropospheric $O_3$ columns that are coherent with the OMI-MLS estimation we used the NCEP tropopause monthly climatology, also available from the GODDARD archive.

## 4 Method

The new $O_3$ reanalysis (hereafter called IASI-r) inherits fundamentally the methodology from IASI-a (Emili et al., 2014; Peiro et al., 2018) but includes the recent developments discussed by Emili et al. (2019) and Aabaribaoune et al. (2020), concerning the direct assimilation of IASI L1 radiances. Therefore, we detail the novelties / changes introduced for this study and address the reader to the previous studies for a more detailed description on the aspects of the methodology that did not change. The reanalysis setup and the differences between IASI-a and IASI-r are also summarised in Table 1.

### 4.1 Model and data assimilation scheme

The CTM MOCAGE is configured on a $2°x2°$ global grid, with 60 vertical levels up to 0.1 hPa. It is forced by ERA-Interim meteorological fields (Dee et al., 2011) and employs a linearised $O_3$ chemistry scheme (Cariolle and Teyssedre, 2007). The CTM configuration is identical to the IASI-a reanalysis (Peiro et al., 2018). As discussed by Emili et al. (2014), the main limitation of this configuration is the missing tropospheric $O_3$ chemistry and the main advantage is the reduced computational cost. It represents, however, a good choice to implement and evaluate the assimilation of new satellite retrievals. Indeed, a similar setup has been used to compute the four years long $O_3$ reanalysis of Van Peet et al. (2018).

The employed assimilation algorithm is a hourly 3D-Var, as in Emili et al. (2019), which differs from the 4D-Var used by Peiro et al. (2018) and Emili et al. (2014). Due to recent optimisations of the MOCAGE code, the former implementation of the linearised and adjoint CTM codes are not yet available within the latest versions of the assimilation suite, which determined the switch to 3D-Var. However, assimilation experiments conducted with MLS observations revealed that $O_3$ differences between a 3D and 4D-Var algorithm are very small within the adopted model configuration (less than 1% difference on global averages, not shown).

The assimilation of IASI L1c spectra is performed through the Radiative Transfer Model (RTM) RTTOV V11.3 (Saunders et al., 2013). The treatment of surface emissivity, skin temperature and all other variables related to the RT follows the method described in Emili et al. (2019). The assimilation of MLS $O_3$ profiles is performed as described either in Peiro et al. (2018) or Emili et al. (2019). To summarize, the CTM is kept identical to IASI-a and the data assimilation changed only in terms of: 4D-Var (IASI-a) versus 3D-Var (IASI-r), treatment of IASI data (L1 radiances assimilated in IASI-r instead of L2 columns in IASI-a) and version of the radiative transfer model being used (RTTOV11 for radiance assimilation instead of RTTOV9 for L2 retrievals assimilation).

### 4.2 Observations

We assimilate only clear-sky IASI L1c radiances in the main $O_3$ window (980-1100 $cm^{-1}$) using the same channel selection as described in Emili et al. (2019). A slightly different clear-sky selection procedure than in Emili et al. (2019) has been employed here since the availability of cloud retrievals from AVHRR and IASI is not homogeneous during 2010: the AVHRR cloud mask is not available before May 2010 in the EUMETSAT L1c data and EUMETSAT L2 cloud retrievals are flawed from September 2010 to December 2010. Therefore, we used the EUMETSAT L2 cloud mask until September 2010 and the

AVHRR cloud mask afterward. In both cases the same strict threshold of 1% is imposed on the cloud fraction to reduce at maximum possible cloud contamination (Emili et al., 2019). Pixels affected by strong dust spectral signature are also filtered out (Emili et al., 2019) as well as those having a surface emissivity smaller than 0.95 (mostly deserts), to avoid potential issues with $O_3$ inversion. Data thinning is performed using a regular grid of 1°×1° resolution and selecting the first pixel that falls in every two grid boxes. This ensures a minimum distance of 1° among assimilated observations. Finally, a dynamical filter is used to reject pixels that differ from modelled radiances by more than 12%. This is done to avoid assimilating observations that, for some undetected reason (e.g. erroneous surface properties or poor model representativity), differ significantly from the model counterparts. The value of the threshold is about twice the standard deviation of the observation minus background values and allows to reject a relatively small number of potential outliers (< 5%, see also Emili et al. (2019)) that might have passed the previous filters. After the selection, the number of assimilated IASI observations varies between 6000 to about 8000 per day.

The single and most significant difference on the assimilation of IASI radiances with respect to Emili et al. (2019) is that we employed a diagnosed observation error covariance ($\mathbf{R}$) instead of the prescribed and diagonal one used previously. This choice is motivated by the results of Aabaribaoune et al. (2020), who found that a diagnosed $\mathbf{R}$ with non-zero inter-channel error correlations can reduce stratospheric biases otherwise introduced by IASI assimilation. The diagnostic of $\mathbf{R}$ is based on innovation statistics (Desroziers et al., 2005) and we followed the procedure suggested by Aabaribaoune et al. (2020): i) we run a first assimilation experiment using the same $\mathbf{R}$ as in Emili et al. (2019) (diagonal with a standard deviation of 0.7 mW m$^{-2}$ sr $^{-1}$ cm$^{-1}$) ii) we diagnosed $\mathbf{R}$ on a average period of one month iii) we used the obtained $\mathbf{R}$ to run a second assimilation experiment for a longer period (12 months) and estimated again $\mathbf{R}$. The latter $\mathbf{R}$ estimation is the one that is used to compute IASI-r because it provides slightly superior results with respect to the first estimation (not shown, see also the discussion in Aabaribaoune et al. (2020)). The employed $\mathbf{R}$ (Fig. 1) is found to be similar to the error covariance matrices estimated by Aabaribaoune et al. (2020), i.e. it presents significant inter-channel error correlations and an error standard deviation matching the typical spectral signature of ozone absorption in the IR.

Concerning MLS, we assimilated only $O_3$ concentrations above 170 hPa and make use of retrieval errors to prescribe $\mathbf{R}$ (Emili et al., 2019).

### 4.3 Background error covariance

The background error covariance $\mathbf{B}$ has been slightly improved with respect to previous studies, which prescribed $O_3$ error profiles based on a single and fixed tropopause height (Emili et al., 2014; Peiro et al., 2018; Emili et al., 2019; Aabaribaoune et al., 2020). For this study the tropopause height is computed every three hours based on the model temperature profiles (ERA-Interim) and using the 2 K m$^{-1}$ lapse rate definition of the World Meteorological Organisation (WMO). The background error standard deviation is prescribed as a percentage of the background $O_3$ profile but with refined values depending on the layer (10% in the troposphere, 1% close to the tropopause, 4% in the lower stratosphere and 2% in the upper stratosphere) and with a correct definition of the tropopause that depends now on the hour and the geographical position. The smaller errors in

correspondence of the tropopause layer reduce the spread of large stratospheric increments into the troposphere. Horizontal and vertical error correlations have been kept the same as in Emili et al. (2019).

Using the new dynamical parameterisation of the background standard deviation instead of the one described by Peiro et al. (2018) provided mostly a reduction of bias of about 10% around the tropopause (not shown).

## 5 Results

We examine $O_3$ simulations for the full year 2010, which is characterised by a sharp transition from positive (El Niño) to negative (La Niña) ENSO conditions (Peiro et al., 2018). The corresponding large departure from climatological $O_3$ in the tropics represents an ideal condition for evaluating chemical simulations. Looking at a full year allows also a complete $O_3$ seasonal cycle in the extra-tropics to be evaluated, which is missing in recent studies (Emili et al., 2019; Aabaribaoune et al., 2020).

The IASI-r analysis has been initialised with the output of a free MOCAGE simulation on 1st January 2010 and run until 31st December 2010, assimilating IASI and MLS observations. In parallel, a free model simulation without data assimilation has been computed for the same period and named Control. Note that the control simulation is identical to the one already discussed in Peiro et al. (2018).

First, IASI-r is validated against ozonesondes and compared to IASI-a (Sec. 5.1), to quantify the improvements with respect to the previous reanalysis. Then, a multi-model inter-comparison is presented (Sec. 5.2, 5.3 and 5.4), to evaluate IASI-r against existent state-of-the-art chemistry model and reanalyses and OMI-MLS retrievals. All reported statistics represent either yearly or monthly data averages.

### 5.1 Validation against ozonesondes

Matches between IASI-r and ozonesondes profiles have been computed online during the simulation, through a linear interpolation of hourly $O_3$ fields at the time and location of each ozonesonde measurement. Original ozonesonde profiles were interpolated to a coarser vertical grid, representative of MOCAGE vertical resolution prior to this matching. All pairs of matched profiles are thus defined over the same pressure grid (using missing values when necessary), which allows a straightforward computation of average statistics. The same strategy has been used to compute matches between ozonesondes and IASI-a, except for the temporal interpolation, which is done using 6-hourly model outputs instead of hourly ones, due to the limited availability of archived IASI-a data. The impact of the different temporal interpolation on the validation statistics has been found to be not significant.

The gain on Root Mean Square Error (RMSE) with respect to ozonesondes for the full year is depicted in Fig. 2 for IASI-r and IASI-a (as percent of the ozonesondes concentration). The gain is computed by means of subtracting the RMSE of IASI-r and IASI-a from that of the control simulation, like in Emili et al. (2019). Negative values in Fig. 2 mean that the $O_3$ RMSE is improved with respect to the control simulation, positive values mean degradation. This type of statistical indicator is commonly used to highlight differences between data assimilation experiments. We remark that both IASI-r and IASI-a

improve the global RMSE by an amount of 5% to 20% depending on the altitude. The two reanalyses show almost identical skills above 300 hPa, due to the positive impact of MLS that is assimilated in both experiments. However, IASI-r shows slightly superior results in the free troposphere (350-800 hPa). Differences in the troposphere become more evident when the RMSE gain is computed for different latitude bands. A well known problem that affected previous reanalyses done using IASI $O_3$

retrievals was the appearance of positive biases in the southern hemisphere and at high latitudes (Emili et al., 2014), which is highlighted by the degradation of RMSE of the IASI-a reanalysis. With IASI-r such degradation is not present: the RMSE gain is neutral in the southern hemisphere and turns to slightly positive in the northern hemisphere. In the tropics, where IASI has the largest sensitivity to tropospheric $O_3$, both IASI-r and IASI-a show the largest improvements of the $O_3$ profile, with similar gains of about 20%, except for a spike of RMSE degradation that still occurred within IASI-a at about 800 hPa.

Pairs of ozonesondes and model instantaneous profiles are used to compute monthly averaged $O_3$ columns, to further focus on the tropospheric $O_3$ variability. Fig. 3 reports free-troposphere (345-750 hPa) $O_3$ columns measured by ozonesondes and the corresponding reanalyses values. The altitude range has been chosen to better investigate differences that appeared in Fig. 2. The figure reports absolute $O_3$ columns (in DU) for ozonesondes, IASI-r, IASI-a and the control simulation plus the corresponding relative differences between the simulation and the ozonesondes (right column). We remark that the seasonal

variability of the $O_3$ column in the extra tropics is much better described by IASI-r than by IASI-a, which is also reflected in the global plot. In particular, the difference between IASI-r and ozonesondes columns remains mostly below 10% whereas it reached values of +30% (-20%) in the 30°S - 60°S (30°N - 60°N) latitude band with IASI-a, sometimes even exceeding the error of the control simulation. The correspondence between IASI-r and ozonesondes is even remarkably good in the Artic circle (60°N - 90°N). Both IASI-r and IASI-a fail to reproduce the seasonal cycle of tropospheric $O_3$ in the Antarctic region

(60°S - 90°S), which was also absent in the control simulation. In the tropical band (30°S - 30°N) IASI-r and IASI-a provide similar results, as already suggested from Fig. 2.

It is highly likely that the large improvements in the extra tropics with respect to IASI-a are a consequence of the direct assimilation of IASI L1 radiances, which represents the main upgrade in the methodology. This interpretation is supported also by the results of Barret et al. (2020), who found a significant improvement in $O_3$ columns in the extra tropics when using a

dynamical a-priori in the retrieval algorithm instead of the constant one used originally Barret et al. (2011). We remind that IASI-a was computed via the assimilation of the $O_3$ retrievals described in Barret et al. (2011) after the application of an empirical bias correction of 10% (Emili et al., 2014; Peiro et al., 2018). Hence, another interesting results is that IASI-r can achieve very similar results to IASI-a in the tropics without any empirical bias correction procedure.

The neutral or negative impact of IASI assimilation in the Antarctic region is probably due to difficulties with the cloud

detection over icy surfaces (Ruston and Mcnally, 2012). Nevertheless, IASI-r $O_3$ fields remain closer to the control simulation than IASI-a, which introduced a small but unrealistic seasonal variability. Overall, we conclude that IASI-r provides a significant improvement over past attempts to assimilate IASI tropospheric $O_3$ products, especially at mid and high latitudes.

## 5.2 Comparison of Tropospheric Ozone Columns

Ozonesondes are valuable measurements to evaluate modeled ozone profiles in the troposphere but their geographical distribution is uneven and their number is relatively small in the tropical and Southern Hemisphere (SH) latitudes. OMI-MLS retrievals (Ziemke et al., 2006, 2011) provide a satellite-based estimation of the TOC that can be used to evaluate models geographical variability in tropical and mid-latitude regions. Hence, OMI-MLS TOC permits to evaluate models in regions that are not well covered by ozonesondes, the main limitation being the lack of vertical information within the troposphere.

Some care must be taken to allow a meaningful comparison between OMI-MLS TOC and the corresponding modeled quantity (Ziemke et al., 2006). The monthly climatology of the tropopause height used within the OMI-MLS algorithm (NCEP model, Ziemke et al. (2011)) has been also employed here to compute tropospheric columns for all the modelling experiments. This was done after the vertical interpolation of modeled $O_3$ fields on a common vertical pressure grid, which has been chosen identical to that available for the CAMSRA database. This approach minimises potential TOC discrepancies due to different tropopause computation or due to different vertical resolutions among models. Monthly TOC fields have been first computed for each model and then averaged temporally to compute TOC maps the four different seasons (DJF, MAM, JJA, SON), displayed in Fig. 4, 5, 6 and 7 respectively.

During DJF months (Fig. 4) the global TOC is the lowest (28.6 DU for OMI-MLS retrievals). IASI-r, IASI-a and CAMSRA show slightly higher TOC values than OMI-MLS but rather similar geographical distribution to OMI-MLS and among each other. The GEOS-CCM model shows larger TOC maxima and lower minima with respect to the other models and OMI-MLS. The MOCAGE control simulation shows less pronounced zonal variability than all other models, which is expected due to the missing tropospheric chemistry description. The visible zonal structures in the control simulation are mostly a result of the zonal variability of the tropopause height.

An increase of TOC is observed in the Northern Hemisphere (NH) mid-latitudes during MAM months (Fig. 5). Values of TOC larger than 40 DU are observed with OMI-MLS over populated continental regions as well as over oceans. Larger TOC values are predicted by CAMSRA and GEOS-CCM compared to MOCAGE based simulations and OMI-MLS. Due to the small sensitivity of IASI IR observations to the boundary layer $O_3$, IASI-r and IASI-a share the same limitations of the control simulation, i.e. a relaxation toward a zonal $O_3$ climatology in the lowermost model layers that is negatively biased (Emili et al., 2014). Such biases do affect the TOC computation to a limited extent. Nevertheless, IASI-r provides slightly larger TOC values than IASI-a, which seems more coherent with full-chemistry models and ozonesondes (Fig. 3). The decrease of TOC in the SH during the MAM months is reproduced by all models.

During JJA period (Fig. 6) values of TOC reach the maximum in the NH mid-latitudes with local maxima in the Mediterranean region, South East Asia and China. As noted previously, GEOS-CCM tends to provide larger TOC variability and extrema. IASI-r and CAMSRA provide the best match to OMI-MLS measurements in this period of the year, whereas IASI-a and the control simulation underestimate the TOC. The increase of TOC in the SH tropical and mid-latitudes with respect to MAM period is reproduced by all models except for the control simulation.

In SON months (Fig. 7) the TOC decreases significantly in the NH but increases in the SH, with a well known local maximum stretching from the South Indian Ocean to the Atlantic Ocean (Liu et al., 2017). IASI-r, CAMSRA and, in a lesser extent IASI-a, provide TOC distributions that match quite well with OMI-MLS measurements. GEOS-CCM simulates a smaller SH maximum and too large TOC values in the NH mid-latitudes. On the other end, the MOCAGE control simulation underestimates TOC values at all latitudes and does not reproduce the expected regional variability.

We remark finally that both IASI-r and IASI-a slightly overestimate TOC values inside the tropical deep convergence zone in South East Asia. This behaviour is observed for all periods of the year.

Overall IASI-r and CAMSRA TOC values are the closest to OMI-MLS measurements, both in terms of regional variability and amplitude, and for most of 2010. IASI-a TOC shows similar global patterns to IASI-r but with reduced amplitude, as anticipated from the previous section (Fig. 3).

## 5.3 Ozone ENSO index

We computed the $O_3$ ENSO index (Ziemke et al., 2011) for the ensemble of the $O_3$ analyses that are discussed in this study and compared to the index derived directly from OMI-MLS observations. The $O_3$ ENSO index is defined as the difference (in DU) between the average tropospheric $O_3$ column in South East Asia (15° S-15° N and 70°-140° E) and in the Pacific Ocean tropical band (15° S-15° N and 180°-110° W). The $O_3$ ENSO index is strongly correlated with ENSO variability and has already been used to evaluate the capacity of CTMs and CCMs to reproduce tropical $O_3$ variability (Ziemke et al., 2015; Peiro et al., 2018).

This comparison does not represent an independent validation for CAMSRA, where both MLS and OMI measurements are assimilated. Nevertheless, it offers a useful benchmark for the other simulations presented in this study.

We report in Fig. 8 the $O_3$ ENSO index for 2010. As expected and already discussed by Peiro et al. (2018), the control simulation is not able to reproduce the $O_3$ ENSO signal observed by OMI-MLS, especially concerning the positive phase of the index. IASI-a has a good variability but slightly negative bias, which was already diagnosed by Peiro et al. (2018) for reasons that were not clear at that time. GEOS-CCM displays smaller biases but a shift of one or two months concerning the transition between the positive and negative ENSO phases. IASI-r shows both small bias and a remarkably good correlation with OMI-MLS. Similar considerations can be given for CAMSRA reanalysis.

To summarize, IASI-r improved the description of tropical $O_3$ variability linked to ENSO with respect to IASI-a and performs very similarly to CAMSRA, which has a full description of tropospheric chemistry and constrains tropospheric $O_3$ thanks to the simultaneous assimilation of both OMI and MLS. The new results also suggest that the bias of IASI-a could be linked to the limits of the empirical bias correction adopted by Emili et al. (2014) and Peiro et al. (2018) when assimilating IASI L2 retrievals. This problem is avoided with the direct assimilation of L1 radiances.

## 5.4 Comparison of ozone monthly fields

We present in Fig. 9 zonal averages of $O_3$ concentration as a function of the altitude and the month. The figure reports the temporal variability of monthly $O_3$ profiles at different latitudes (from top to bottom) for CAMSRA (first column) plus the

difference between IASI-r, IASI-a, GEOS-CCM, MOCAGE control simulation and CAMSRA (from left to right). CAMSRA, which is based on both comprehensive tropospheric chemistry and satellite assimilation, is considered here as the reference to be matched by the other systems.

In the stratosphere (10-150 hPa) the differences between CAMSRA and IASI-r are smaller than 5%, except close to the tropical tropopause, where they top at about 20%. On the other hand, the differences between CAMSRA and our control simulation are significantly larger at all latitudes (up to 50%). We can conclude that the strong similarities of stratospheric $O_3$ between CAMSRA and IASI-r are a consequence of assimilating MLS profiles in both reanalyses, and not of employing the same chemical mechanism (Cariolle and Teyssèdre, 2007). The differences at the tropical tropopause might be linked to the different vertical resolution or assimilation configuration, which are expected to play a more significant role where $O_3$ vertical gradients are strong and concentrations are small. The GEOS-CCM simulation, which employs a detailed stratospheric $O_3$ chemical mechanism but does not assimilate any data, shows a persistent negative bias with respect to CAMSRA or IASI-r, of about 5-10 %. Note that the occurence of the South Pole $O_3$ depletion starting in September is significantly underestimated in the control simulation, but also by GEOS-CCM.

In the free troposphere (250-750 hPa) the control simulation has a predominant negative bias (up to 30%) in the tropics and in the NH, whereas biases change sign as a function of the altitude and the month in the SH. These biases are in accordance with previous studies (Emili et al., 2014) and Fig. 3 and are a consequence of the missing tropospheric mechanisms in the linearised chemical scheme. GEOS-CCM shows smaller differences with CAMSRA but systematic negative biases are found in the tropics (10-15%). In the NH a positive bias initially localised in the upper troposphere (30-40% between 150-400 hPa) spreads to the whole troposphere later in the year. Differences between IASI-r and CAMSRA are smaller at almost all latitudes with respect to the control simulation and GEOS-CCM, except in the lowermost layers (below 750 hPa), where IASI measurements do not provide sensitivity. Absolute differences with CAMSRA are lower than 10% at all latitudes and seasons, except in the SH and springtime period.

IASI-a results are reported for completeness (Fig. 9, third column) and their comparison with IASI-r (second column) is coherent with previous findings (Sec. 5.1): IASI-r reduced significantly the tropospheric biases of IASI-a at mid and high-latitudes, both in the SH and NH. Absolute differences between IASI-r/IASI-a and CAMSRA remain instead of the same order (10%) in the tropics but differ in sign as a function of the altitude and the month.

To summarise, the monthly variability of $O_3$ profiles provided by IASI-r is the closest to that of CAMSRA. Since the assimilation of MLS measurements does not influence the $O_3$ profiles below 300 hPa (Emili et al., 2014, 2019) we can conclude that the significant reduction of tropospheric biases with respect to the control simulation during the entire year is a consequence of IASI assimilation. Thanks to the good coverage of IASI measurements, IASI-r also provides smaller differences with CAMSRA than GEOS-CCM, which employs a comprehensive chemical mechanism but is not constrained by satellite observations in this study.

## 6 Conclusions

The objective of this study was to re-evaluate the impact of IASI IR measurements on global tropospheric $O_3$ analysis, after recent improvements in methodological aspects concerning the data assimilation scheme (Emili et al., 2019; Aabaribaoune et al., 2020). In particular, we recomputed the $O_3$ reanalysis of Peiro et al. (2018) for the year 2010 but assimilating directly IR radiances from IASI instead of Level 2 retrievals. We named the new reanalysis IASI-r as opposed to IASI-a of Peiro et al. (2018), which shows little or no benefit from IASI assimilation in the extra-tropics. Most aspects of the system (chemical transport model, assimilation of stratospheric $O_3$ from MLS) were kept unchanged with respect to IASI-a, to evaluate mostly the impact of the radiances assimilation. Some adjustments of the background error covariance have been also performed, which, however, produced a minor impact on reanalysis results.

IASI-r was first validated against ozonesondes and the scores compared with those of IASI-a. The scores of IASI-r and IASI-a are similar at tropical latitudes but are significantly better for IASI-r in the extra-tropics. The assimilation of IASI improves now the RMSE of the control simulation both in the SH mid-latitudes and in the NH polar region, whereas IASI-a was found to degrade the RMSE at these latitudes. The monthly variability of tropospheric $O_3$ columns confirms these findings and demonstrate that some positive seasonal signal can clearly be extracted from IASI at most latitudes. The only region where the impact is still neutral is between 60°S - 90°S, probably because of the difficulties of cloud screening and $O_3$ retrieval over elevated and icy surfaces.

Further analysis of IASI-r/IASI-a and two state-of-the-art modelling and data assimilation products (GEOS-CCM and CASMRA) was conducted in the tropical region, by comparing the $O_3$ ENSO index. We found that IASI-r provides, together with CAMSRA, values of the ENSO index that are the closest to well established satellite estimates (OMI-MLS) and reduces significantly the negative bias of IASI-a. We remind that, while IASI-a did already capture quite well the $O_3$ variability in the tropics, an empirical bias correction of Level 2 retrievals was necessary to avoid residual biases in the reanalysis (Emili et al., 2014). Residual biases in the $O_3$ index demonstrated the difficulties with such approach. Within IASI-r, no observational bias correction has been performed and this was not found to be detrimental for the reanalysis. Barret et al. (2020) showed that the $O_3$ prior information used in L2 retrievals was likely a reason for some of the tropospheric biases found previously with IASI assimilation. The differences found between IASI-r and IASI-a in this study point to the same conclusion, i.e. that a dynamical $O_3$ prior information, which comes directly from the CTM forecasts in our case, is beneficial for the assimilation of IR measurements. It might be interesting in the future to evaluate the assimilation of improved Level 2 retrievals (Barret et al., 2020) and see wether this also solves the issues encountered previously with IASI-a.

Finally, the temporal variability of vertical $O_3$ profiles from IASI-r, IASI-a, GEOS-CCM and the MOCAGE control simulation (linearised $O_3$ chemistry) has been compared to that provided by the CAMSRA reanalysis, which has been extensively validated in other studies and proven to be quite accurate. The monthly $O_3$ profiles of IASI-r are the closest to CAMSRA, confirming that a simple chemical mechanism in combination with the assimilation of both stratospheric and tropospheric $O_3$ sounders can provide similar performances of more complex setups. Thus, the methodology presented in this study could be useful to improve the tropospheric $O_3$ description in NWP systems (usually based on simplified $O_3$ chemistry), where IR

measurements are already assimilated to constrain temperature and water vapour profiles. The additional cost of the RTM computations would in this case scale linearly with the number of added $O_3$ sensitive channels. In this work we used the original spectral selection of Barret et al. (2011) but the presence of strong inter-channel error correlations (Aabaribaoune et al., 2020) suggests the possibility to reduce significantly the number of assimilated channels without degrading the analysis. This represents an interesting area for further research.

The GEOS-CCM simulation, which is based on comprehensive chemical mechanism but does not assimilate any satellite in this study, sits between our control simulation and IASI-r. Hence, we advocate that assimilating IR measurements from IASI might be beneficial also for models that are based on more detailed $O_3$ chemistry. An evaluation of the added value of IASI assimilation within the CAMSRA system would also offer further insights on the relative importance of UV and IR sounders for tropospheric $O_3$ reanalyses.

A strong data thinning of IASI observations has been performed in this study to match the low resolution of the CTM ($2°$), which resulted in rejecting many potentially informative IASI pixels. The small numerical cost of the CTM configuration permits an increase in horizontal resolution and thus the possibility to use significantly more satellite observations. A strong increase in the number of assimilated pixels might require to investigate on the presence and the potential implementation of spatially correlated observation errors, which are still neglected in most systems. Further improvements in IR $O_3$ assimilation might also come from simulating the effect of large aerosols particles (e.g. dust) in the radiative transfer computations. All these aspects deserve to be considered in future studies.

*Code and data availability.* The model simulations presented in this study (IASI-r, IASI-a and control simulation) as well as the input data (observations, meteorological fields and error covariance matrices) are available through the permanent link https://b2share.cerfacs.fr/records/e3493962fb0e4e6b918d846b070de9d9 . The software used to generate the analysis (MOCAGE and its variational assimilation suite, release version 2020.1.0) is a research-operational code property of Météo France and CERFACS and is not publicly available. The readers interested in obtaining parts of the code for research purposes can contact directly the authors of this study.

*Author contributions.* M. El Aabaribaoune provided the computer code to compute the IASI error covariance matrix and revised the manuscript. E. Emili carried out the experiments, analysed the results and and wrote the manuscript.

*Competing interests.* The authors declare that they have no conflict of interest.

*Acknowledgements.* We acknowledge EUMETSAT for providing IASI L1C data, WOUDC for providing ozonesondes data, the NASA Jet Propulsion Laboratory for the Aura MLS Level 2 $O_3$ used in this study. We thank the Copernicus team at ECMWF for providing the CAMSRA renalysis and the Centre for Environmental Data Analysis (CEDA) for providing GEOS-CCM data from the ACCMIP project.

We also thank the MOCAGE team at Météo-France for providing the chemical transport model, the RTTOV team for the radiative transfer model, J. Ziemke and the NASA Goddard Space Flight Center for the OMI-MLS tropospheric $O_3$ data. We finally acknowledge Andrea Piacentini and Gabriel Jonville for the help on technical developments of the MOCAGE assimilation code. This work has been possible thanks to the financial support from the Région Midi-Pyrénées, who sponsored the preliminary work of Hélène Peiro on the subject, and CNES (Centre National d'Études Spatiales), through the TOSCA program.

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

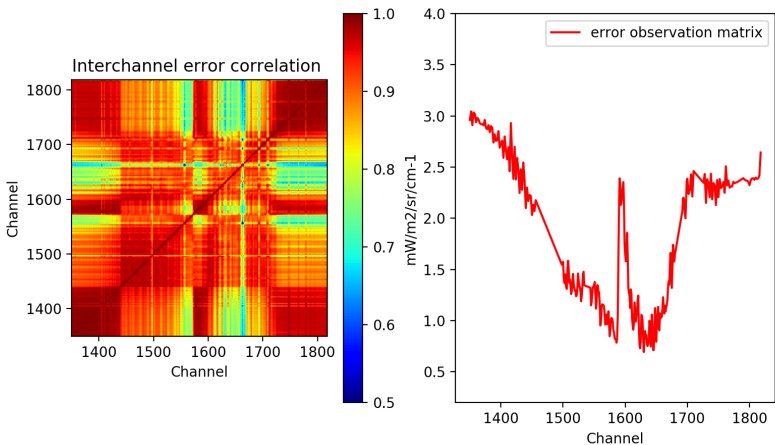

**Figure 1.** Observation error covariance used to assimilate IASI L1 radiances in this study: correlation matrix (left plot) and error standard deviation (squared diagonal of the covariance matrix, right plot). The X-Y axes on the left plot and the X axis on the right plot report the number of IASI spectral channel within the 980-1100 $cm^{-1}$ window.

M., Zeng, G., and Ziemke, J.: Tropospheric Ozone Assessment Report (TOAR): Assessment of global-scale model performance for global and regional ozone distributions, variability, and trends, Elementa: Science of the Anthropocene, 6, 0–84, https://doi.org/10.1525/elementa.265, http://eprints.lancs.ac.uk/88836/1/TOAR{_}Model{_}Performance{_}07062017.pdf{%}0Ahttp://www.igacproject.org/sites/default/files/2017-05/TOAR-Model{_}Performance{_}draft{_}for{_}open{_}comment.pdf, 2017.

Ziemke, J. R., Chandra, S., Duncan, B. N., Froidevaux, L., Bhartia, P. K., Levelt, P. F., and Waters, J. W.: Tropospheric ozone determined from Aura OMI and MLS: Evaluation of measurements and comparison with the Global Modeling Initiative's Chemical Transport Model, Journal of Geophysical Research, 111, D19 303, https://doi.org/10.1029/2006JD007089, http://dx.doi.org/10.1029/2006JD007089, 2006.

Ziemke, J. R., Chandra, S., Labow, G. J., Bhartia, P. K., Froidevaux, L., and Witte, J. C.: A global climatology of tropospheric and stratospheric ozone derived from Aura OMI and MLS measurements, Atmospheric Chemistry and Physics, 11, 9237–9251,

https://doi.org/10.5194/acp-11-9237-2011, http://www.atmos-chem-phys.net/11/9237/2011/, 2011.

Ziemke, J. R., Douglass, A. R., Oman, L. D., Strahan, S. E., and Duncan, B. N.: Tropospheric ozone variability in the tropics from ENSO to MJO and shorter timescales, Atmospheric Chemistry and Physics, 15, 8037–8049, https://doi.org/10.5194/acp-15-8037-2015, 2015.

Ziemke, J. R., Oman, L. D., Strode, S. A., Douglass, A. R., Olsen, M. A., McPeters, R. D., Bhartia, P. K., Froidevaux, L., Labow, G. J., Witte, J. C., Thompson, A. M., Haffner, D. P., Kramarova, N. A., Frith, S. M., Huang, L. K., Jaross, G. R., Seftor, C. J., Deland, M. T., and Taylor,

S. L.: Trends in global tropospheric ozone inferred from a composite record of TOMS/OMI/MLS/OMPS satellite measurements and the MERRA-2 GMI simulation, Atmospheric Chemistry and Physics, 19, 3257–3269, https://doi.org/10.5194/acp-19-3257-2019, 2019.

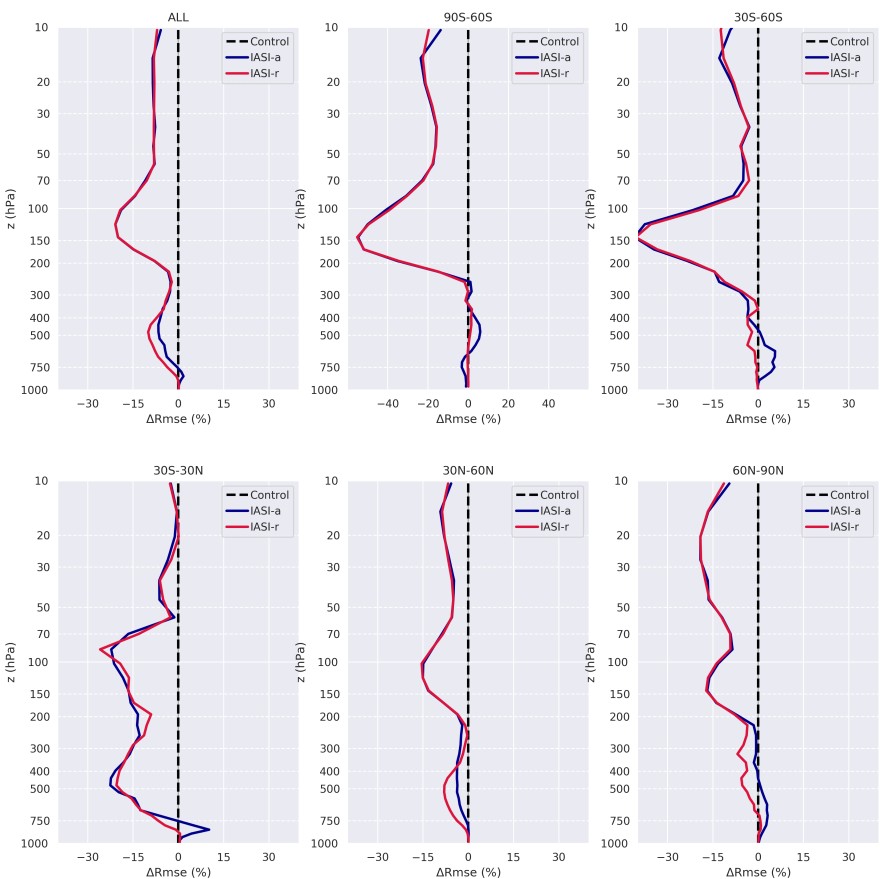

**Figure 2.** Relative gain of Root Mean Square Error (RMSE) with respect to radiosoundings averaged globally (first plot) and for five latitude bands separately (90°S-60°S, 60°S-30°S, 30°S-30°N, 30°N-60°N, 60°N-90°N). IASI-a is depicted in blue and IASI-r in red. Negative values indicate a decrease (improvement) of RMSE with respect to the control simulation (dotted line), positive values indicate an increase (degradation) of RMSE.

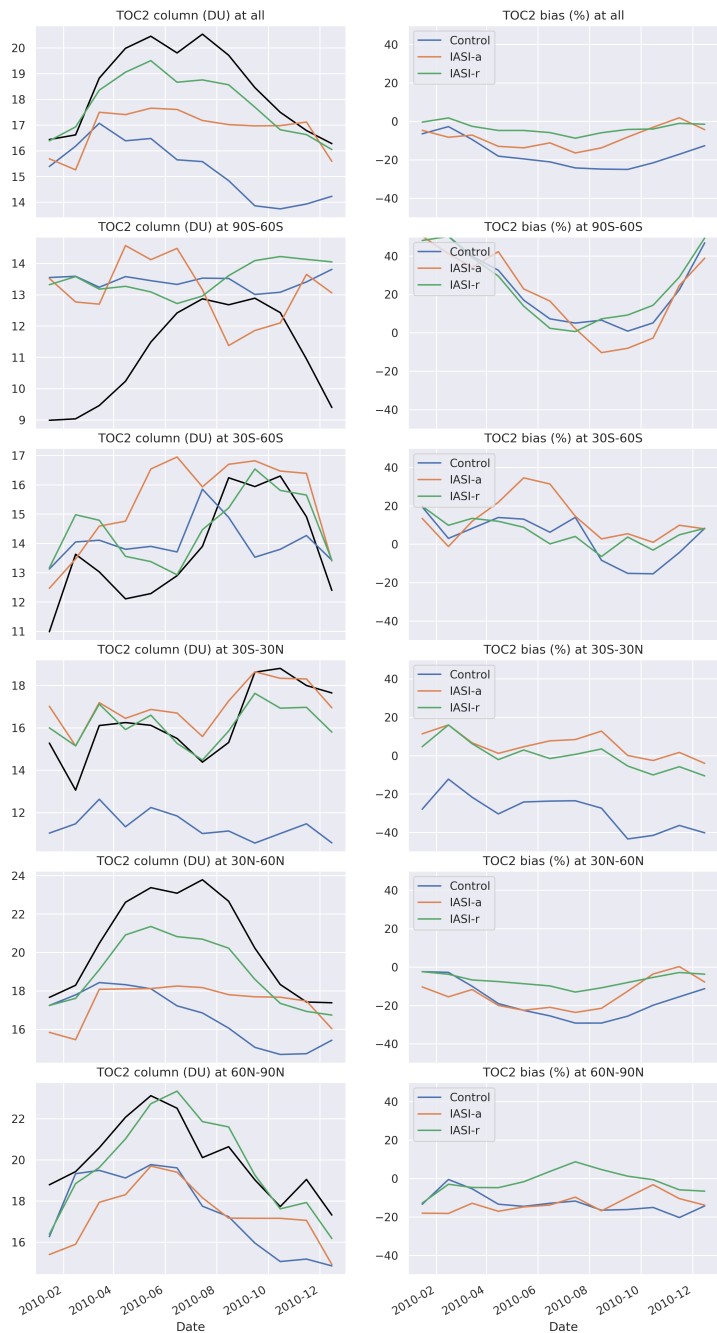

**Figure 3.** Monthly variability of the ozone free troposphere column (345-750 hPa) averaged globally (first line) and for five latitude bands separately (90°S-60°S, 60°S-30°S, 30°S-30°N, 30°N-60°N, 60°N-90°N). On the left: absolute values of O$_3$ columns (in DU) for IASI-a (orange line), IASI-r (green line), the control simulation (blue line) and corresponding ozonesondes columns (black line). On the right: relative differences between simulated and corresponding ozonesonde columns (in percent of ozonesonde columns).

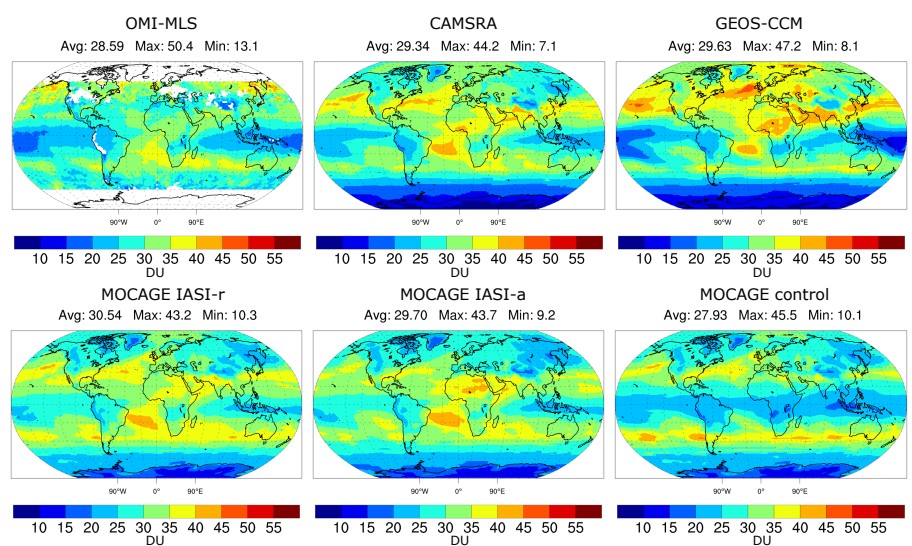

**Figure 4.** Tropospheric Ozone Column averaged on December-January-February 2010 from: OMI-MLS retrievals (top left), CAMSRA reanalysis (top middle), GEOS-CCM simulation (top right), IASI-r reanalysis (bottom left), IASI-a reanalysis (bottom middle) and MOCAGE control simulation (bottom right). Mean, minimum and maximum values of ozone (in DU) are depicted over each plot.

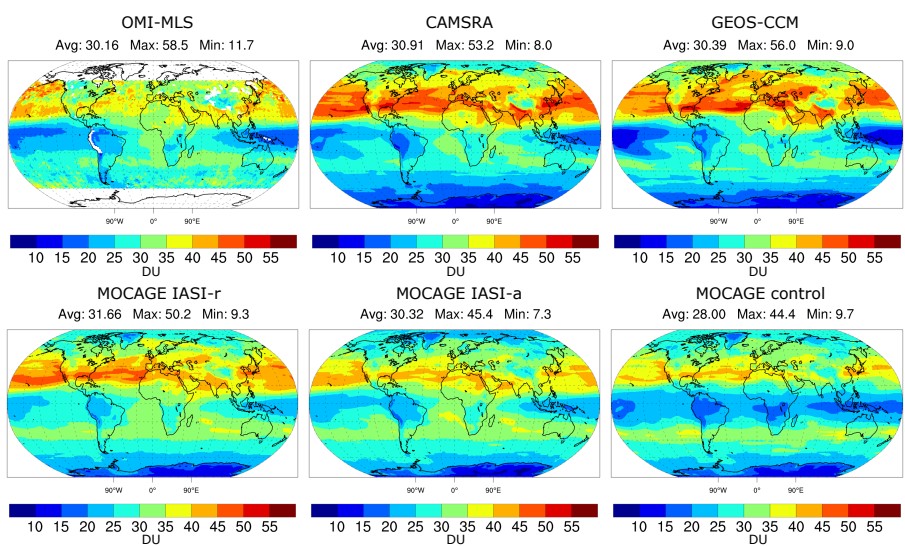

**Figure 5.** Tropospheric Ozone Column averaged on March-April-May 2010 from: OMI-MLS retrievals (top left), CAMSRA reanalysis (top middle), GEOS-CCM simulation (top right), IASI-r reanalysis (bottom left), IASI-a reanalysis (bottom middle) and MOCAGE control simulation (bottom right). Mean, minimum and maximum values of ozone (in DU) are depicted over each plot.

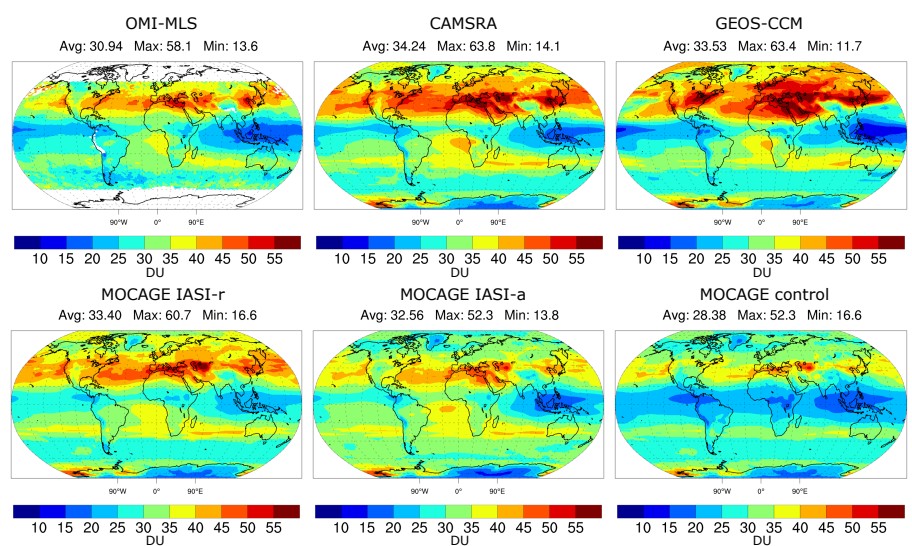

**Figure 6.** Tropospheric Ozone Column averaged on June-July-August 2010 from: OMI-MLS retrievals (top left), CAMSRA reanalysis (top middle), GEOS-CCM simulation (top right), IASI-r reanalysis (bottom left), IASI-a reanalysis (bottom middle) and MOCAGE control simulation (bottom right). Mean, minimum and maximum values of ozone (in DU) are depicted over each plot.

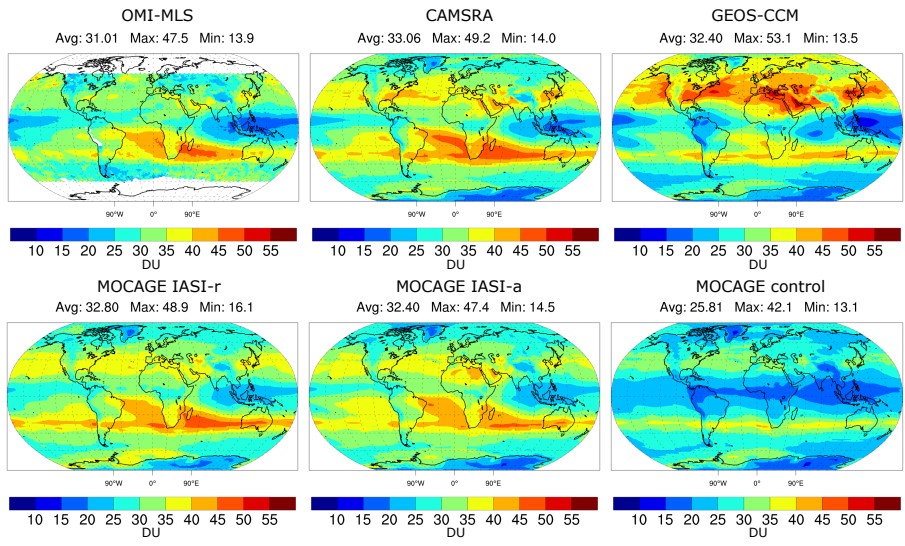

**Figure 7.** Tropospheric Ozone Column averaged on September-October-November 2010 from: OMI-MLS retrievals (top left), CAMSRA reanalysis (top middle), GEOS-CCM simulation (top right), IASI-r reanalysis (bottom left), IASI-a reanalysis (bottom middle) and MOCAGE control simulation (bottom right). Mean, minimum and maximum values of ozone (in DU) are depicted over each plot.

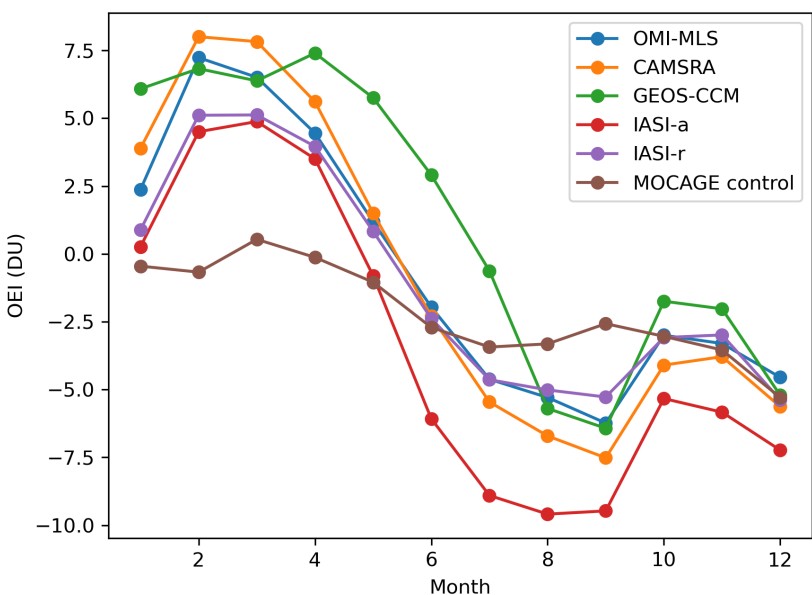

**Figure 8.** Ozone Enso Index (OEI) from: OMI-MLS measurements (blue line), CAMSRA reanalysis (orange), GEOS-CCM simulation (green), IASI-a reanalysis (red), IASI-r reanalysis (violet) and MOCAGE control simulation (brown).

**Table 1.** Configuration of IASI-r and IASI-a (Peiro et al., 2018) experiments. The dash symbol indicates that the configuration is the same. * AK stands for Averaging Kernels.

|  | IASI-a | IASI-r |
| --- | --- | --- |
| CTM resolution | 2°x2°, 60 levels (surface-0.1 hPa) | - |
| Meteorological forcing | ERA-Interim | - |
| Chemical scheme | Linear $O_3$ chemistry | - |
| Assimilation Algorithm | 4D-Var (12 hours window) | 3D-Var (1 hour window) |
| IASI assimilation | L2 columns (1000-345 hPa) with AK* | L1 radiances (980 - 1100 $cm^{-1}$) |
| IASI measurements errors | 15% plus empirical bias correction of -10% | Estimated covariance matrix |
| MLS assimilation | MLS v4.2 $O_3$ profiles | - |
| Background standard deviation | 15% troposphere, 5% stratosphere | 10% troposphere, 2-4% stratosphere |
| Tropopause height (for background error) | Constant (100 hPa) | Local (based on T profile) |
| Horizontal error correlation scale | 500 km | 200 km |
| Vertical error correlation scale | Zero (no correlation) | 1 grid point |

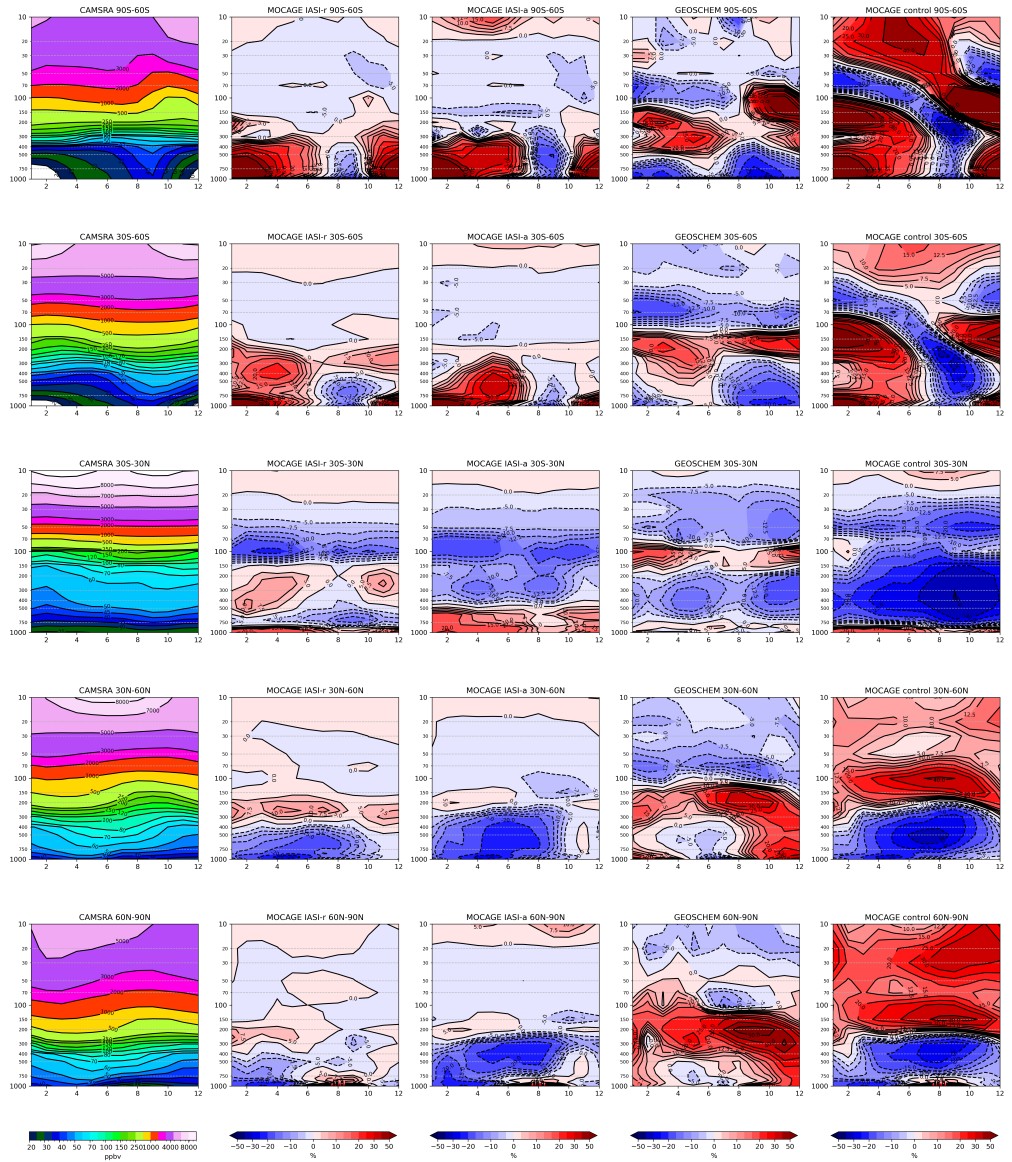

**Figure 9.** Ozone zonal averages as a function of month (x-axis) and altitude (y-axis in hPa) for five latitude bands separately (90°S-60°S, 60°S-30°S, 30°S-30°N, 30°N-60°N, 60°N-90°N from top to bottom). CAMSRA O$_3$ (in ppbv) is plotted on the first column. The relative differences between IASI-r (second column), IASI-a (third column), GEOS-CCM (fourth column), MOCAGE control simulation (fifth column) and CAMSRA O$_3$ are given in percent of the CAMSRA O$_3$.