# Peer review of "Impact of IASI thermal infrared measurements on global ozone reanalyses"

_Geoscientific Model Development, 2020_

## Author Comment (AC1)

**Impact of IASI thermal infrared measurements on global ozone reanalyses. Reply to referee # 1**

Emanuele Emili[1] and Mohammad El Aabaribaoune[1]

[1]CECI, Université de Toulouse, Cerfacs, CNRS, Toulouse, France

**Correspondence:** Emili (emili@cerfacs.fr)

**1 Reply to general comments**

We would like to thank the reviewer for his comments, which helped to improve significantly the manuscript. Detailed replies to his comments follow:

1. *The authors now mostly focus on evaluations of zonal mean concentration at five latitude bands (Figs. 2.3), or as zonal mean (Figure 5). In addition it would be interesting to see the performance of the assimilation system for more regional variations, e.g. by plotting presenting evaluations as in Figs 2/3 specific for certain regions (e.g. Europe), or presenting the data as in Figure 5, but (for instance) showing a lot/lat plot averaged over DJF and JJA, at selected altitude levels.*

   **Answer**: We included in the revised manuscript global maps of Tropospheric Ozone Column (1000 hPa - tropopause), as defined in Ziemke et al. (2006), averaged on DJF, MAM, JJA and SON months (Fig. 1, 2, 3, 4 in this document respectively). These plots allow now to evaluate the geographical distribution of tropospheric ozone from the different models with respect to OMI-MLS retrievals. The same tropopause level (NCEP climatology) used to compute OMI-MLS retrievals have been used to compute the model columns, as already done previously to compute Fig. 4 of the original manuscript. A new section has been added in the revised manuscript (after Sec. 5.1) to discuss the new figures, which is reported below:

   **Comparison of tropospheric ozone columns**

[revised manuscript text omitted]

2. *In Figure 5, in addition to showing the results of IASI-r, why not also present plots for the IASI-a product for reference?*

**Answer**:

The IASI-a plots were omitted because IASI-a was already validated against ozone-sondes and found to be less accurate than IASI-r (Sec. 5.1 of the original manuscript). This choice was also taken to reduce the figure's size and improve its readability. Following the comment of the reviewer, we decided to include IASI-a plots (see Fig. 5) so that all simulations are now displayed in the model vertical inter-comparison section. The following text has been added in the manuscript to comment IASI-a plots:

IASI-a results are reported for completeness (Fig. 5, third column) and their comparison with IASI-r (second column) confirms the previous findings (Sec. 5.1): IASI-r reduced significantly the tropospheric biases of IASI-a at mid and high-latitudes both in the SH and NH. Absolute differences between IASI-r/IASI-a and CAMSRA remain instead of the same order (10%) in the tropics but differ in sign as a function of the altitude and the month.

3. *Can the authors comment on the differences in computational costs for the L1 and L2 assimilation configurations, and discuss the reasons for differences, and potential means to improve on this? Also, it's good to put these differences*

[Figure]

**Figure 2.** Tropospheric Ozone Column averaged on March-April-May 2010 from: OMI-MLS retrievals (top left), CAMSRA reanalysis (top middle), GEOS-CCM simulation (top right), IASI-r reanalysis (bottom left), IASI-a reanalysis (bottom middle) and MOCAGE control simulation (bottom right). Mean, minimum and maximum values of ozone (in DU) are depicted over each plot.

[Figure]

**Figure 3.** Tropospheric Ozone Column averaged on June-July-August 2010 from: OMI-MLS retrievals (top left), CAMSRA reanalysis (top middle), GEOS-CCM simulation (top right), IASI-r reanalysis (bottom left), IASI-a reanalysis (bottom middle) and MOCAGE control simulation (bottom right). Mean, minimum and maximum values of ozone (in DU) are depicted over each plot.

[Figure]

**Figure 4.** Tropospheric Ozone Column averaged on September-October-November 2010 from: OMI-MLS retrievals (top left), CAMSRA reanalysis (top middle), GEOS-CCM simulation (top right), IASI-r reanalysis (bottom left), IASI-a reanalysis (bottom middle) and MOCAGE control simulation (bottom right). Mean, minimum and maximum values of ozone (in DU) are depicted over each plot.

*in costs into perspective, particularly in regard to their comment in the conclusion in the case "IR measurements are already assimilated", (p12, l16)*

**Answer**: A first analysis of the difference in computational cost between L1 and L2 assimilation was given in Emili et al. (2019) Sec. 4.2. The numerical cost of assimilating L1 data is of course larger than for L2 (by a factor of 3.5 in the former study). However, the former analysis only compared the cost of the 3D-Var assimilation itself, without counting the numerical cost of the L2 retrievals. L2 retrievals also involve Radiative Transfer Model (RTM) computations and, in the particular case of the 1D-Var retrievals assimilated in IASI-a, multiple calls to the linearised and adjoint RTM are also performed during the minimisation. Considering the contribution of L2 retrievals in the overall cost will reduce significantly the differences between the two approaches. For a given number of satellite observations and using the same RTM code for both L1 assimilation and L2 retrievals (RTTOV in our case), the computational difference between the 'L2 retrieval plus assimilation' and 'L1 assimilation' can only arise from the number of iterations needed to reach convergence. The number of needed iterations for L2 retrievals and L1 assimilation depends on many factors (choice of the minimiser, scene/pixel, other assimilated instruments etc.) and it is fine tuned for each application. In general, a larger number of iterations of the minimiser is needed to find 3D or 4D-Var solutions with respect to a collection of 1D retrievals solutions. For example, about 30 iterations are needed in average for IASI L1 + MLS L2 3D-Var assimilation (IASI-r). Hence we expect that L1 assimilation might remain overall more expensive than L2 retrievals plus assimilation. A precise quantification of this overhead would require computing again the L2 retrievals with the same version of the

[Figure]

**Figure 5.** Ozone zonal averages as a function of month (x-axis) and altitude (y-axis in hPa) for five latitude bands separately (90°S-60°S, 60°S-30°S, 30°S-30°N, 30°N-60°N, 60°N-90°N from top to bottom). CAMSRA O$_3$ (in ppbv) is plotted on the first column. The relative differences between IASI-r (second column), IASI-a (third column), GEOS-CCM (fourth column), MOCAGE control simulation (fifth column) and CAMSRA O$_3$ are given in percent of the CAMSRA O$_3$.

RTM used in this study (RTTOV 11) and on the same CPUs of the L1 assimilation. This would demand a significant amount of work. However, it can be argued that this overhead would become of lesser importance when some O$_3$ IR

channels are already used for NWP (Dragani and Mcnally, 2013). In this case only the cost of including additional $O_3$ channels has to be considered. Moreover, we used until now the same IASI channel selection of Barret et al. (2011) for L1 assimilation, which seem to provide some redundant information. Ongoing studies are focused on the reduction of needed channels in the $O_3$ band, which represent the most straightforward way to reduce the computational cost of L1 assimilation (and L2 retrievals as well).

Since this discussion is quite technical and the experiments needed to give a precise quantification of the computational overhead of L1 assimilation cannot be done easily we prefer not to include it in the revised manuscript. However, we added the following text in the conclusions (p12, l16) to elaborate a bit further:

The additional cost of the RTM computations would in this case scale linearly with the number of added $O_3$ sensitive channels. In this work we used the original spectral selection of Barret et al. (2011) but the presence of strong inter-channel error correlations (Aabaribaoune et al., 2020) suggests the possibility to reduce significantly the number of assimilated channels without degrading the analysis. This represents an interesting area for further research.

4. *Furthermore, considering the data-thinning approach (pp 7, line 3), in the current study only a single observation is used in every 2x2 grid box. Can the authors explain a bit more about their data-thinning approach, e.g. do you make any additional check on selecting representative observations (apart from the dynamic filter)?*

**Answer**: The data thinning is performed as follows in practice: we used a regular grid of $1° \times 1°$ resolution and select the first pixel that falls in every two grid boxes. This ensures a minimum distance of $1°$ among assimilated observations. No additional criteria have been used to screen observations based on representativeness because it is not straightforward to define such criteria in the radiance space. We think that increasing the model resolution in the future constitutes the best way to exploit the large number of IASI observations.

The following sentence has been added in the revised manuscript for sake of clarity:

Data thinning is performed using a regular grid of $1° \times 1°$ resolution and selecting the first pixel that falls in every two grid boxes. This ensures a minimum distance of $1°$ among assimilated observations.

**2 Reply to specific comments**

**Answer**:

All specific comments have been integrated in the revised manuscript.

---

## Author Comment (AC2)

**Impact of IASI thermal infrared measurements on global ozone reanalyses. Reply to referee # 2**

Emanuele Emili[1] and Mohammad El Aabaribaoune[1]

[1]CECI, Université de Toulouse, Cerfacs, CNRS, Toulouse, France

**Correspondence:** Emili (emili@cerfacs.fr)

**1   Reply to specific comments**

We thank the anonymous reviewer for his comments, which helped to improve significantly the manuscript. Detailed replies to his comments follow:

1. *page 2 lines 15-20: the uninterrupted nature of IASI measurements versus the interrupted nature of TES measurements is a key motivation for the paper. However because the reader is not told what measurements Inness et al. (2019) use, this motivation is not so clear.*

   **Answer**: The fact that IASI measurements were not assimilated in Inness et al. (2019) was written at page 2 line 11-12. However, for better clarity we also modified the last sentence of the paragraph as follows:

   Although IASI $O_3$ observations might be very valuable for long reanalyses, they have not yet been assimilated within any of the currently available chemical reanalysis.

2. *page 6 line 18: it would be beneficial to quantify 'very small', even just a headline number to give the reader a feel for the significance of changing from 4D- to 3D-Var*

   **Answer**: The information has been added to the text as follows:

   However, assimilation experiments conducted with MLS observations revealed that $O_3$ differences between a 3D and 4D-Var algorithm are very small within the adopted model configuration (less than 1% difference on global averages, not shown).

3. *page 6 line 19: as pointed out by the authors the radiative transfer model changes between IASI-a and IASI-r, but there is no comment on the effect this has on resulting O3 fields. The reader would appreciate the distinction between the effect of changing IASI product level and the effect of changing radiative transfer model.*

   **Answer**: We agree with the reviewer about the fact that the impact of every single incremental change in the analysis system should be quantified and analysed. However, this is not always possible due to the significant time needed to compute and analyse satellite retrievals and data assimilation experiments. The changes between L2 and L1 assimilation

using the same version of the RTM (RTTOV v11) has been thoroughly discussed in Emili et al. (2019) and the reference given in the introduction (page 2 lines 29-30). The impact of only changing the RTM version demands to recompute the L2 retrievals and re-assimilate them. This is unfortunately not possible in the framework of this study since the L2 $O_3$ retrievals are produced by an external entity and require significant computational and human resources. We already mentioned in the conclusions the potential interest of recomputing the IASI-a analysis based on a new version of the SOFRID L2 algorithm (Barret et al., 2020), to check wether the most recent version of L2 products might give similar results to IASI-r (page 12 lines 9-10). However, this is out of scope for the present manuscript, whose main purpose is to evaluate the overall impact of all the modifications introduced within the assimilation scheme since the publication of IASI-a (page 3 lines 8-10).

4. *page 7 line 3: could the reason for using a dynamical filter to reject pixels that differ from the model by more than 12% be given?*

**Answer**: The dynamical filter constitutes the last resort to discard observations that were initially considered for assimilation but differ substantially from the model counterpart. This step is needed because a careful selection of observations based only on cloud/dust contamination and surface emissivity is not always sufficient to exclude some satellite pixels that can still degrade the analysis. This can happen for example due to miss-predicted snow surfaces, undetected clouds/aerosols, poor representativity due to the model-satellite resolution mismatch. The threshold of 12% has been tuned based on observations minus background histograms computed for some days of assimilation. The purpose was to exclude the observations that fall outside the $2\sigma$ confidence interval. We report in Fig. 1 the geographical distribution of the background minus observation values and the effect of the emissivity and dynamical filters during the first 5 days of the IASI-r analysis, for a subset of assimilated channels. We can remark that all spectral channels show similar behaviour, with a standard deviation ranging from about 4% to 7% of observed radiances (histograms in Fig. 1). Also highest discrepancies are generally observed over land (top maps in Fig. 1), where surface emissivity values are lower and more uncertain. As a consequence of our filtering strategy, most of the rejected observations (brown pixels in the bottom plot) are located over land. The total number of rejected observations is about 12% of the initially retained ones for this 5 days period. We cannot display separately the effects of surface emissivity and dynamical filters here because we did not store the results of the action of each single filter in our DA system. Emili et al. (2019) used the same threshold of 12% for the dynamical filter and observed a ratio of about 3% of rejection due to this filter alone in July 2010 (no surface emissivity filter was needed in their study).

The following sentence has been included in the manuscript to resume this discussion:

Finally, a dynamical filter is used to reject pixels that differ from modelled radiances by more than 12%. This is done to avoid assimilating observations that, for some undetected reason (e.g. erroneous surface properties or poor model representativity), differ significantly from the model counterparts. The value of the threshold is about twice the standard deviation of the observation minus background values and allows to reject a relatively small number of potential outliers (< 5%, see also Emili et al. (2019)) that might have passed the previous filters.

[Figure]

**Figure 1.** Background minus observations values (as % of observed values) for 6 IASI channels within the O$_3$ band cumulated during the first 5 days of IASI-r. On top: maps of all individual satellite pixels before the application of the surface emissivity and dynamical filters. In the middle: frequency histograms corresponding to the pixels above. Each map / histogram corresponds to a single IASI channel, whose number is indicated on top of the plot. On the bottom: map of selected (grey) and rejected (brown) pixels after the joint application of the surface emissivity and dynamic filters.

**2 Reply to technical comments**

**Answer**:

All technical comments have been integrated in the revised manuscript.

**References**

Barret, B., Emili, E., and Le Flochmoen, E.: A tropopause-based a priori for IASI-SOFRID Ozone retrievals: improvements and validation, Atmospheric Measurement Techniques Discussions, 2020, 1–35, https://doi.org/10.5194/amt-2020-5, https://amt.copernicus.org/preprints/amt-2020-5/, 2020.

5 Emili, E., Barret, B., Le Flochmoën, E., and Cariolle, D.: Comparison between the assimilation of IASI Level 2 ozone retrievals and Level 1 radiances in a chemical transport model, Atmospheric Measurement Techniques, 12, 3963–3984, https://doi.org/10.5194/amt-12-3963-2019, https://www.atmos-meas-tech.net/12/3963/2019/, 2019.